# DETERMINISTIC POLICY IMITATION GRADIENT ALGORITHM

## ABSTRACT

The goal of imitation learning (IL) is to enable a learner to imitate an expert's behavior given the expert's demonstrations. Recently, generative adversarial imitation learning (GAIL) has successfully achieved it even on complex continuous control tasks. However, GAIL requires a huge number of interactions with environment during training. We believe that IL algorithm could be more applicable to the real-world environments if the number of interactions could be reduced. To this end, we propose a model free, off-policy IL algorithm for continuous control. The keys of our algorithm are two folds: 1) adopting deterministic policy that allows us to derive a novel type of policy gradient which we call deterministic policy imitation gradient (DPIG), 2) introducing a function which we call state screening function (SSF) to avoid noisy policy updates with states that are not typical of those appeared on the expert's demonstrations. Experimental results show that our algorithm can achieve the goal of IL with at least tens of times less interactions than GAIL on a variety of continuous control tasks.

## 1 INTRODUCTION

Recent advances in reinforcement learning (RL) have achieved super-human performance on several domains (Mnih et al., 2015; Silver et al., 2016; Mnih et al., 2016; Lillicrap et al., 2015). However, on most of domains with the success of RL, the design of reward, that explains what agent's behavior is favorable, is clear enough for humans. Conversely, on domains where it is unclear how to design the reward, agents trained by RL algorithms often obtain poor policies and their behavior is far from what we want them to do. Imitation learning (IL) comes in such cases. The goal of IL is to enable the learner to imitate the expert's behavior given the expert's demonstrations but the reward signals. We are interested in IL because we desire an algorithm that can be applied in real-world environments where it is often hard to design the reward. Besides, since it is generally hard to model a variety of the real-world environments with an algorithm, and the state-action pairs in a vast majority of the real-world applications such as robotics control can be naturally represented in continuous spaces, we focus on model free IL for continuous control.

A widely used approach of existing model free IL methods is the combination of Inverse Reinforcement Learning (IRL) (Russell, 1998; Ng et al., 2000; Abbeel & Ng, 2004; Ziebart et al., 2008) and RL. Recently, Ho & Ermon (2016) has proposed generative adversarial imitation learning (GAIL) on the line of those works. GAIL has achieved state-of-the art performance on a variety of continuous control tasks. However, as pointed out by Ho & Ermon (2016), a crucial drawback of GAIL is requirement of a huge number of interactions between the learner and the environments during training[1]. Since the interactions with environment can be too much time-consuming especially in the real-world environments, we believe that model free IL could be more applicable to the real-world environments if the number could be reduced while keeping the imitation capability satisfied as well as GAIL.

To reduce the number of interactions, we propose a model free, off-policy IL algorithm for continuous control. As opposed to GAIL and its variants (Baram et al., 2017; Wang et al., 2017; Hausman et al., 2017; Li et al., 2017) those of which adopt a stochastic policy as the learner's policy, we adopt a

---

[1]Throughout this paper, we define "the number of interactions" as he number of state-action pairs obtained through the interactions with environment during training, but those obtained in expert's demonstrations.

deterministic policy while following adversarial training fashion as GAIL. We show that combining the deterministic policy into the adversarial off-policy IL objective derives a novel type of policy gradient which we call deterministic policy *imitation* gradient (DPIG). Because DPIG only integrates over the state space as deterministic policy gradient (DPG) algorithms (Silver et al., 2014), the number of the interactions is expected to be less than that for stochastic policy gradient (PG) which integrates over the state-action space. Besides, we introduce a function which we call *state screening function* (SSF) to avoid noisy policy update with states that are not typical of those appeared on the experts demonstrations.

In order to evaluate our algorithm, we used 6 physics-based control tasks that were simulated with MuJoCo physics simulator (Todorov et al., 2012). The experimental results show that our algorithm enables the learner to achieve the same performance as the expert does with at least tens of times less interactions than GAIL. It indicates that our algorithm is more applicable to the real-world environments than GAIL.

## 2 BACKGROUND

### 2.1 PRELIMINARIES

We consider a Markov Decision Process (MDP) which is defined as a tuple $\{\mathcal{S}, \mathcal{A}, \mathcal{T}, r, d_0, \gamma\}$, where $\mathcal{S}$ is a set of states, $\mathcal{A}$ is a set of possible actions agents can take, $\mathcal{T} : \mathcal{S} \times \mathcal{A} \times \mathcal{S} \to [0, 1]$ is a transition probability, $r : \mathcal{S} \times \mathcal{A} \to \mathbb{R}$ is a reward function, $d_0 : \mathcal{S} \to [0, 1]$ is a distribution over initial states, and $\gamma \in (0, 1]$ is a discount factor. The agent's behavior is defined by a stochastic policy $\pi : \mathcal{S} \times \mathcal{A} \to [0, 1]$. Performance measure of the policy is defined as $\mathcal{J}(\pi, r) = \mathbb{E}\left[\sum_{t=0}^{\infty} \gamma^t r(s_t, a_t) | d_0, \mathcal{T}, \pi\right]$ where $s_t \in \mathcal{S}$ is a state that the agent received at discrete time-step $t$, and $a_t \in \mathcal{A}$ is an action taken by the agent after receiving $s_t$. Using discounted state visitation distribution (SVD) denoted by $\rho_\pi(s) = \sum_{t=0}^{\infty} \gamma^t \mathbb{P}(s_t = s | d_0, \mathcal{T}, \pi)$ where $\mathbb{P}$ is a probability that the agent receives the state $s$ at time-step $t$, the performance measure can be rewritten as $\mathcal{J}(\pi, r) = \mathbb{E}_{s \sim \rho_\pi, a \sim \pi}\left[r(s, a)\right]$. In IL literature, the agent indicates both the expert and the learner. In this paper, we consider that the states and the actions are represented in continuous spaces $\mathcal{S} = \mathbb{R}^{d_s}$ and $\mathcal{A} = \mathbb{R}^{d_a}$ respectively.

### 2.2 REINFORCEMENT LEARNING

The goal of RL is to find an optimal policy that maximizes the performance measure. In this paper, we consider the policy-based methods rather than the value-based method such as Q-learning. Given the reward function $r$, the objective of RL with parameterized stochastic policies $\pi_\theta$ is defined as follows.

$$\mathrm{RL}(r) = \mathrm{argmax}_\theta \ \mathcal{J}(\pi_\theta, r) = \mathrm{argmin}_\theta \ -\mathcal{J}(\pi_\theta, r) \tag{1}$$

Typically, as do REINFORCE (Williams, 1992) family of algorithms, the update of $\theta$ is performed by gradient ascent with estimations of $\nabla_\theta \mathcal{J}(\pi_\theta, r)$ which is called PG (Sutton et al., 2000). There are several different expressions of PG which can be represented as follows.

$$\nabla_\theta \mathcal{J}(\pi_\theta, r) = \mathbb{E}_{\zeta \sim \pi_\theta}\left[\sum_{t=0}^{\infty} \Psi_t(s_t, a_t) \nabla_\theta \log \pi_\theta(a_t | s_t)\right] \tag{2}$$

Where $\zeta$ denotes trajectories generated by the agent with current policy $\pi_\theta$, and $\Psi_t(s_t, a_t)$ can be chosen from a set of the expressions for expected discounted cumulative rewards $\mathbb{E}\left[\sum_{t'=t}^{\infty} \gamma^{t'-t} r(s_{t'}, a_{t'}) | s_t, a_t\right]$. We refer the readers to (Schulman et al., 2015b) for the expressions in detail. As mentioned in Section 1, we consider deterministic policy $\mu_\theta : \mathcal{S} \to \mathcal{A}$ which is a special case of stochastic policy. In the RL literature, gradient estimation of the performance measure for deterministic policy which is called DPG (Silver et al., 2014) can be represented as follows.

$$\nabla_\theta \mathcal{J}(\mu_\theta, r) = \mathbb{E}_{s \sim \rho_{\mu_\theta}}[\nabla_a Q(s, a)|_{a=\mu_\theta(s)} \nabla_\theta \mu_\theta(s)] \tag{3}$$

Where $Q : \mathcal{S} \times \mathcal{A} \to \mathbb{R}$ denotes Q-function. Note that, both PG and DPG require calculation or approximation of the expected discounted cumulative rewards, and do not suppose that the reward function is accessible. That is, as opposed to IRL algorithms, RL algorithms using PG or DPG do not suppose that the reward function is parameterized.

### 2.3 Inverse Reinforcement Learning

The goal of IRL is to find a reasonable reward function based on an assumption that the cumulative rewards earned by the expert's behavior are greater than or equal to those earned by any non-experts' behavior. The objective of IRL with parameterized reward functions $r_\omega$ can be defined as follows.

$$\text{IRL}(\pi_E) = \text{argmax}_\omega \ \mathcal{J}(\pi_E, r_\omega) - \mathcal{J}(\pi, r_\omega) \tag{4}$$

Where $\pi_E$ and $\pi$ denote the expert's and the non-expert's policies respectively. The update of $\omega$ is typically performed by gradient ascent with estimations of $\nabla_\omega \mathcal{J}(\pi_E, r_\omega) - \nabla_\omega \mathcal{J}(\pi, r_\omega)$.

### 2.4 Imitation Learning

Likewise IRL the reward function is unknown in IL setting. Hence, the decision process turns to be characterized by MDP\$r = \{\mathcal{S}, \mathcal{A}, \mathcal{T}, d_0, \gamma\}$ and the performance measure becomes unclear unless the reward function can be found. Given the parameterized reward functions $r_\omega$ learned by IRL, we suppose the decision process becomes MDP $\cup \ r_\omega \setminus r = \{\mathcal{S}, \mathcal{A}, \mathcal{T}, r_\omega, d_0, \gamma\}$. We consider the widely used IL approach that combines IRL and RL, and its objective can be defined as a composition of (1) and (4) as follows.

$$\text{IL}(\pi_E) = \text{RL} \circ \text{IRL}(\pi_E) = \text{argmin}_\theta \ \text{argmax}_\omega \ \mathcal{J}(\pi_E, r_\omega) - \mathcal{J}(\pi_\theta, r_\omega) \tag{5}$$

It is common to alternate IRL and RL using the gradient estimation of (5) with respect to $\omega$ and to $\theta$ respectively. We can see this IL process as adversarial training as pointed out in (Ho & Ermon, 2016; Finn et al., 2016a).

## 3 Algorithm

### 3.1 Deterministic Policy Imitation Gradient

We define the parameterized reward functions as $r_\omega(s, a) = \log R_\omega(s, a)$, where $R_\omega : \mathcal{S} \times \mathcal{A} \to [0, 1]$. $R_\omega(s, a)$ represents a probability that the state-action pair $(s, a)$ belong to the trajectories demonstrated by the expert, and thus $r_\omega(s, a)$ represents its log probability. Besides, we introduce a stochastic behavior policy (Precup et al., 2006; Degris et al., 2012) $\beta : \mathcal{S} \times \mathcal{A} \to [0, 1]$ and its discounted SVD $\rho_\beta(s) = \sum_{t=0}^\infty \gamma^t \mathbb{P}(s_t = s | d_0, \mathcal{T}, \beta)$. We thereby define the objective of IRL in our algorithm as follows.

$$\text{argmax}_\omega \ \mathbb{E}_{s \sim \rho_{\pi_E}, a \sim \pi_E}[\log R_\omega(s, a)] + \mathbb{E}_{s \sim \rho_\beta, a \sim \beta}[\log(1 - R_\omega(s, a))] \tag{6}$$

Although we adopt deterministic policy $\mu_\theta : \mathcal{S} \to \mathcal{A}$ as mentioned in Section 1, a well-known concern for the deterministic policy is about its exploration in the state space. To ensure the adequate exploration with the deterministic policy, we introduce an off-policy learning scheme with the stochastic behavior policy $\beta$ introduced above. We assume that the stochastic behavior policy $\beta$ is a mixture of the past learner's policies, and approximate the performance measure for $\mu_\theta$ as $\mathcal{J}_\beta(\mu_\theta, r_\omega) \approx \mathbb{E}_{s \sim \rho_\beta}[r_\omega(s, \mu_\theta(s))]$ as off-policy DPG algorithms (Silver et al., 2014). Therefore, the objective of the learner to imitate the expert's behavior can be approximated as follows.

$$\text{argmax}_\theta \ \mathbb{E}_{s \sim \rho_\beta}[\log R_\omega(s, \mu_\theta(s))] \tag{7}$$

We alternate the optimization for (6) and (7) as GAIL. In practice, as do common off-policy RL methods (Mnih et al., 2015; Lillicrap et al., 2015), we perform the state sampling $s \sim \rho_\beta$ using a replay buffer $\mathcal{B}_\beta$. The replay buffer is a finite cache and stores the state-action pairs in first-in first-out manner while the learner interacts with the environment.

Whereas RL algorithms using DPG do not suppose that the reward function is parameterized as mentioned in Section 2.2, the reward function in (7) is parameterized. Hence, our algorithm can obtain gradients of the reward function with respect to action executed by the deterministic policy, and apply the chain rule for the gradient estimation of (7) with respect to $\theta$ as follows.

$$\nabla_\theta \mathcal{J}_\beta(\mu_\theta, r_\omega) \approx \mathbb{E}_{s \sim \rho_\beta}[\nabla_a \log R_\omega(s, a)|_{a=\mu_\theta(s)} \nabla_\theta \mu_\theta(s)] \tag{8}$$

Thus, combining the deterministic policy into the adversarial IL objective with the parameterized reward function enables us to derive this novel gradient estimation (8) which we call deterministic policy *imitation* gradient (DPIG).

### 3.2 State Screening Function

Let $\mathcal{S}_\beta \subset \mathcal{S}$ and $\mathcal{S}_E \subset \mathcal{S}$ be subsets of $\mathcal{S}$ explored by the learner and the expert respectively, let $\mathcal{S}_{\beta*E} = \mathcal{S}_\beta \cap \mathcal{S}_E$ and $\overline{\mathcal{S}_{\beta*E}} = \mathcal{S} \setminus \mathcal{S}_{\beta*E}$ be intersection of the two subsets and its complement, and let $U_\omega : \mathcal{S} \to \Delta\theta$ be updates of the learner's policy with DPIG. If there exist states $s \in \mathcal{S}_{\beta*E}$, we would say the mapping $U_\omega(s)$ for the states is reasonable to get $r_\omega(s, \mu_\theta(s))$ close to $r_\omega(s, a)$ where $a \sim \pi_E(\cdot|s)$. In other words, since $r_\omega$ can reason desirable actions $a \sim \pi_E(\cdot|s)$ and undesirable actions $\mu_\theta(s)$ for the states $s \in \mathcal{S}_{\beta*E}$ through comparisons between $(s, a)$ and $(s, \mu_\theta(s))$ in IRL process, $U_\omega(s)$ for the states could reasonably guide the leaner to right directions to imitate the expert's behavior. Conversely, if there exist states $s \in \overline{\mathcal{S}_{\beta*E}}$, the mapping $U_\omega(s)$ for the states would be unreasonable, since the comparisons for the states are not performed in IRL process. As a result, $U_\omega(s)$ for the states $s \in \overline{\mathcal{S}_{\beta*E}}$, which we call *noisy policy updates*, often guide the leaner to somewhere wrong without any supports. In IL setting, $\mathcal{S}_{\beta*E}$ is typically small due to a limited number of demonstrations, and $\overline{\mathcal{S}_{\beta*E}}$ is greatly lager than $\mathcal{S}_{\beta*E}$. Thus, the noisy policy updates could frequently be performed in IL and make the learner's policy poor. From this observation, we assume that preventing the noisy policy updates with states that are not typical of those appeared on the expert's demonstrations benefits to the imitation.

Based on the assumption above, we introduce an additional function $v_\eta : \mathcal{S} \to \mathbb{R}$, which we call *state screening function* (SSF), parameterized by $\eta$. We empirically found that SSF works with form $v_\eta(s) = 1/(-\log V_\eta(s) + \epsilon)$ where $V_\eta : \mathcal{S} \to [0, 1]$ and $\epsilon > 0$. $V_\eta(s)$ represents a probability that the state $s$ belongs to $\mathcal{S}_E$. Hence, the values of $v_\eta(s)$ are expected to be much smaller values for the states $s \in \overline{\mathcal{S}_{\beta*E}} \cap \mathcal{S}_\beta$ than those for $s \in \mathcal{S}_{\beta*E} \subseteq \mathcal{S}_E$. We show below the final objective functions $\mathcal{K}_r(\omega)$, $\mathcal{K}_v(\eta)$, and $\mathcal{K}_\mu(\theta)$ to be maximized in our algorithm.

$$\mathcal{K}_r(\omega) = \mathbb{E}_{s \sim \rho_{\pi_E}, a \sim \pi_E}[\log R_\omega(s, a)] + \mathbb{E}_{s \sim \rho_\beta, a \sim \beta}[\log (1 - R_\omega(s, a))] \tag{9}$$

$$\mathcal{K}_v(\eta) = \mathbb{E}_{s \sim \rho_{\pi_E}}[\log V_\eta(s)] + \mathbb{E}_{s \sim \rho_\beta}[\log (1 - V_\eta(s))] \tag{10}$$

$$\mathcal{K}_\mu(\theta) = \mathbb{E}_{s \sim \rho_\beta}[v_\eta(s) \log R_\omega(s, \mu_\theta(s))] \tag{11}$$

The update of parameters $\omega$, $\eta$ and $\theta$ uses the gradient estimations $\nabla_\omega \mathcal{K}_r(\omega)$, $\nabla_\eta \mathcal{K}_v(\eta)$ and $\nabla_\theta \mathcal{K}_\mu(\theta)$ respectively, where $\nabla_\theta \mathcal{K}_\mu(\theta)$ follows DPIG (7). The overview of our algorithm is described in Algorithm 1. Note that, the parameter $\theta$ is updated while accessing states $s \in \mathcal{S}_\beta = (\overline{\mathcal{S}_{\beta*E}} \cap \mathcal{S}_\beta) \cup \mathcal{S}_{\beta*E}$, and SSF works as weighted sampling coefficients of which amounts for the states in $\mathcal{S}_{\beta*E}$ are greater than those for the states in $\overline{\mathcal{S}_{\beta*E}} \cap \mathcal{S}_\beta$. Thus, effects of the noisy policy updates to the learner's policy can be reduced.

One may think that the application of SSF in (11) makes policy updates more similar to applying generative adversarial training over the actions while sampling states from $\rho_E$ as true distribution. However, a notable difference of IL from generative adversarial training such as GANs (Goodfellow et al., 2014) is that sampling states from the SVDs of current learner's policy or behavioral policy are essential for estimating policy gradients that improve the learner's behavior. Whereas, sampling states from distributions, such as $\rho_E$ or uniform distribution on $\mathcal{S}$, would not make sense for the improvement theoretically. We think that sampling states from $\rho_\beta$ with SSF in equation (11) can also be interpreted as sampling states from SVDs of current learner's policy only on $\mathcal{S}_{\beta*E}$ if $V_\eta$ was well approximated.

## 4 Experiments

### 4.1 Setup

In order to evaluate our algorithm, we used 6 physics-based control tasks that were simulated with MuJoCo physics simulator (Todorov et al., 2012). We trained agents on each task by an RL algorithm, namely trust region policy optimization (TRPO) (Schulman et al., 2015a), using the rewards defined in the OpenAI Gym (Brockman et al., 2016), then we used the resulting agents as the experts for IL algorithms. Using a variety number of trajectories generated by the expert's policy as datasets, we trained learners by IL algorithms. See Appendix A for description of each task, an agents performance with random policy, and the performance of the experts. The performance of the experts and the learners were measured by cumulative reward they earned in an episode. To measure how well the learner imitates expert's behavior, we introduced a criterion which we call *performance recovery rate*

---

**Algorithm 1** Overview of our algorithm

---

1: Initialize parameters $\omega$, $\eta$, and $\theta$.
2: Fulfill a $\mathcal{B}_E$ by the state-action pairs from the expert's demonstrations.
3: Initialize replay buffer $\mathcal{B}_\beta \leftarrow \emptyset$.
4: **for** episode = 1, M **do**
5:     Receive initial state $s_0$
6:     Initialize time-step $t = 0$
7:     **while not** terminate condition **do**
8:         Select an action $a_t = \mu_\theta(s_t)$ according to the current policy.
9:         Execute the action $a_t$ and observe new state $s_{t+1}$
10:         Store a state-action pair $(s_{t+1}, a_{t+1})$ in $\mathcal{B}_\beta$.
11:         Sample a random mini-batch of N state-action pairs $(s_i, a_i)$ from $\mathcal{B}_E$.
12:         Sample a random mini-batch of N state-action pairs $(s_j, a_j)$ from $\mathcal{B}_\beta$.
13:         Update $\omega$ and $\eta$ using the sampled gradients $\nabla_\omega \mathcal{K}_r(\omega)$ and $\nabla_\omega \mathcal{K}_r(\omega)$ respectively.
14:             $\nabla_\omega \mathcal{K}_r(\omega) \approx \frac{1}{N} \sum_{i=1}^{N} \nabla_\omega \log R_\omega(s_i, a_i) + \frac{1}{N} \sum_{j=1}^{N} \nabla_\omega \log(1 - R_\omega(s_j, a_j))$
15:             $\nabla_\eta \mathcal{K}_v(\eta) \approx \frac{1}{N} \sum_{i=1}^{N} \nabla_\eta \log V_\eta(s_i) + \frac{1}{N} \sum_{j=1}^{N} \nabla_\eta \log(1 - V_\eta(s_j))$
16:         Sample a random mini-batch of N states $s_k$ from $\mathcal{B}_\beta$.
17:         Update $\theta$ using the sampled gradient $\nabla_\theta \mathcal{K}_\mu(\theta)$:
18:             $\nabla_\theta \mathcal{K}_\mu(\theta) \approx \frac{1}{N} \sum_{k=1}^{N} v_\eta(s_k) \nabla_a \log R_\omega(s_k, a)|_{a=\mu_\theta(s_k)} \nabla_\theta \mu_\theta(s_k)$
19:         $t = t + 1$
20:     **end while**
21: **end for**

---

(PRR). PRR was calculated by $(C - C_0)/(C_E - C_0)$ where $C$ denotes the performance of the learner, $C_0$ denotes the performance of agents with random policies, and $C_E$ denotes the performance of the experts. We tested our algorithm against four algorithms: behavioral cloning (BC) (Pomerleau, 1991), GAIL, MGAIL which is model based extension of GAIL, Ours\SSF which denotes our algorithm without using SSF ($v_\eta$ is always one), and Ours\int which denotes our algorithm sampling states from $\rho_E$ instead of $\rho_\beta$ without using SSF in (11). We ran three experiments on each task, and measure the PRR during training the learner.

We implemented our algorithm with three neural networks with two hidden layers. Each network represented the functions $\mu_\theta$, $R_\omega$, and $V_\eta$. All neural networks were implemented using TensorFlow (Abadi et al., 2016). See Appendix B for the implementation details. For GAIL and MGAIL, we used publicly available code which are provided the authors [2][3]. Note that number of hidden layers and number of hidden units in neural network for representing policy are the same over all methods tested. All experiments were run on a PC with a 3.30 GHz Intel Core i7-5820k Processor, a GeForce GTX Titan GPU, and 32GB of RAM.

## 4.2 RESULTS

Figure 1 shows curves of PRR measured during training with BC, GAIL, Ours\SSF, and Ours\int. The PRR of our algorithm and GAIL achieved were nearly 1.0 over all tasks. It indicates that our algorithm enables the learner to achieve the same performance as the expert does over a variety of tasks as well as GAIL. We also observe that the number of interactions our algorithm required to achieve the PPR was at least tens of times less than those for GAIL over all tasks. It show that our algorithm is much more sample efficient in terms of the number of interactions than GAIL, while keeping imitation capability satisfied as well as GAIL. Table 1 shows that comparison of CPU-time required to achieve the expert's performance between our algorithm and GAIL. We see that our algorithm is more sample efficient even in terms of computational cost.

We observe that SSF yielded better performance over all tasks compared to Ours\SSF. The difference of the PRR is obvious for Ant-v1 and Humanoid-v1 of which state spaces are comparatively greater than those of the other tasks. We also observe that SSF made difference with relatively smaller number of trajectories given as dataset in Reacher-v1 and HalfCheetah-v1. It shows that, as we noted in Section 3.2, SSF is able to reduce the effect of the noisy policy updates in the case where $\overline{\mathcal{S}_{\beta * E}}$ is greatly lager than $\mathcal{S}_{\beta * E}$.

---

[2] https://github.com/openai/imitation
[3] https://github.com/itaicaspi/mgail

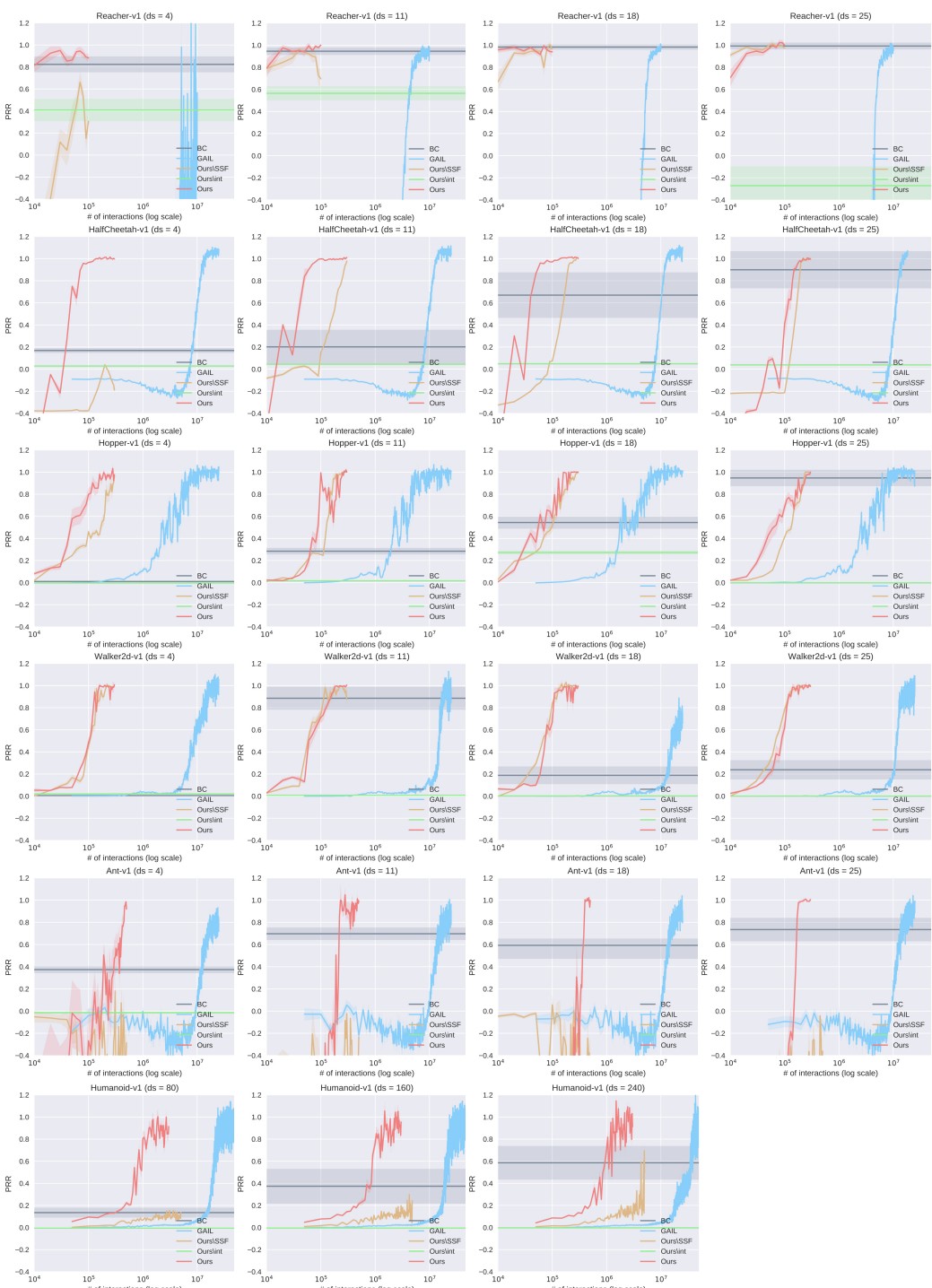

Figure 1: Comparison of the experimental results among BC, GAIL, Ours\SSF, Ours\int, and our algorithm (Ours). The y-axis represents PRR and the x-axis represents the number of interactions with environment during training in log scale. Each row represents different task (in order from top: Reacher-v1, HalfCheetah-v1, Hopper-v1, Walker2d-v1, Ant-v1, and Humanoid-v1). Each column represents different number of the expert's trajectories given as dataset (ds denotes the number).

Table 1: Summary of how much times more CPU-time GAIL required to achieve the expert's performance than that of our algorithm. The number of the expert's trajectories given as dataset is denoted by ds. The CPU-time was measured when satisfying either two criteria – (a) PRR achieved to be greater than or equals to 1.0. (b) PRR marked the highest number.

| Task | ds = 4 | ds = 11 | ds = 18 | Ds = 25 |
|---|---|---|---|---|
| Reacher-v1 | 28.95 | 10.13 | 11.31 | 28.47 |
| HalfCheetah-v1 | 4.40 | 6.71 | 4.65 | 3.74 |
| Hopper-v1 | 4.29 | 1.93 | 3.05 | 2.21 |
| Walker2d-v1 | 9.88 | 6.39 | 14.17 | 8.21 |
| Ant-v1 | 5.47 | 11.81 | 6.41 | 11.34 |
| | ds = 80 | ds = 160 | ds = 240 | |
| Humanoid-v1 | 4.78 | 2.97 | 6.55 | |



Figure 2: Comparison of the experimental results between MGAIL and our algorithm (Ours) on Hopper-v1 task.

Over all tasks, both BC and Ours\int performed worse than our algorithm. It indicates that our algorithm is less affected by the compounding error as described in Section 5, and sampling states from any distribution other than $\rho_\beta$ does not make sense for improving the learner's performance as mentioned in Section 3.2.

Figure 2 depicts the experimental results comparing our algorithm with MGAIL on Hopper-v1[4]. As mentioned by Baram et al. (2017), the number of the interactions MGAIL required to achieve the expert's performance was relatively less than GAIL. However, our algorithm is more sample efficient in terms of the number of interactions than MGAIL.

## 5 RELATED WORK AND DISCUSSION

A wide variety of IL methods have been proposed in these last few decades. The simplest IL method among those is BC (Pomerleau, 1991) which learns a mapping from states to actions in the expert's demonstrations using supervised learning. Since the learner with the mapping learned by BC does not interacts with the environments, inaccuracies of the mapping are never corrected once the training has done, whereas our algorithm corrects the learner's behavior through the interactions. A noticeable point in common between BC and our algorithm is that the both just consider the relationship between single time-step state-action pairs but information over entire trajectories of the behavior in the optimizations. In other words, the both assume that the reward structure is dense and the reasonable rewards for the states $s \in \mathcal{S} \setminus \mathcal{S}_E$ can not be defined. A drawback of BC due to ignorance of the information over the trajectories is referred to as the problem of *compounding error* (Ross & Bagnell, 2010) – the inaccuracies compounds over time and can lead the learner to encounter unseen states in th expert's demonstrations. For the state $s \in \mathcal{S}_{\beta*E}$, it is assumed in our algorithm that the immediate reward is greater if the learner's behavior is more likely to the expert's behavior, and the expert's behavior yields the greatest cumulative reward. That is, maximizing the immediate reward for the

---

[4]We've never achieved the same performance as reported in the paper on all tasks except for Hopper-v1 even if we followed the author's advices.

state $s \in \mathcal{S}_{\beta*E} \subset \mathcal{S}_E$ implies maximizing the cumulative reward for trajectories stating from the state in our algorithm, and thus the information over the trajectories is implicitly incorporated in $\log R_\omega$ of the objective (11). Therefore, our algorithm is less affected by the compounding error than BC.

Another widely used approach for IL the combination of IRL and RL that we considered in this paper. The concept of IRL was originally proposed by Russell (1998), and a variety of IRL algorithms have been proposed so far. Early works on IRL (Ng et al., 2000; Abbeel & Ng, 2004; Ziebart et al., 2008) represented the parameterized reward function as a linear combination of hand-designed features. Thus, its capabilities to represent the reward were limited in comparison with that of nonlinear functional representation. Indeed, applications of the early works were often limited in small discrete domains. The early works were extended to algorithms that enable to learn nonlinear functions for representing the reward (Levine et al., 2011; Finn et al., 2016b). A variety of complex tasks including continuous control in the real-world environment have succeeded with the nonlinear functional representation (Finn et al., 2016b). As well as those methods, our algorithm can utilize the nonlinear functions if it is differentiable with respect to the action.

In recent years, the connection between GANs and the IL approach has been pointed out (Ho & Ermon, 2016; Finn et al., 2016a). Ho & Ermon (2016) showed that IRL is a dual problem of RL which can be deemed as a problem to match the learner's occupancy measure (Syed et al., 2008) to that of the expert, and found a choice of regularizer for the cost function yields an objective which is analogous to that of GANs. After that, their algorithm, namely GAIL, has become a popular choice for IL and some extensions of GAIL have been proposed (Baram et al., 2017; Wang et al., 2017; Hausman et al., 2017; Li et al., 2017). However, those extensions have never addressed reducing the number of interactions during training whereas we address it, and our algorithm significantly reduce the number while keeping the imitation capability as well as GAIL.

The way of deriving policy gradients using gradients of the parameterized reward function with respect to actions executed by the current learner's policy is similar to DPG (Silver et al., 2014) and deep DPG (Lillicrap et al., 2015). However, they require parameterized Q-function approximator with known reward function whereas our algorithm does not use Q-function besides the parameterized reward function learned by IRL. In IL literature, MGAIL (Baram et al., 2017) uses the gradients derived from parameterized discriminator to update the learner's policy. However, MGAIL is model-based method and requires to train parameterized forward-model to derive the gradients whereas our algorithm is model free. Although model based methods have been thought to need less environment interactions than model free methods in general, the experimental results showed that MGAIL needs more interactions than our model free algorithm. We think that the reasons are the need for training the forward model besides that for the policy, and lack of care for the noisy policy updates issue that MGAIL essentially has.

## 6 CONCLUSION

In this paper, we proposed a model free, off-policy IL algorithm for continuous control. The experimental results showed that our algorithm enables the learner to achieve the same performance as the expert does with several tens of times less interactions than GAIL.

Although we implemented shallow neural networks to represent the parameterized functions in the experiment, deep neural networks can also be applied to represent the functions in our algorithm. We expect that the that advanced techniques used in deep GANs enable us to apply our algorithm to more complex tasks.

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

## A    DETAILED DESCRIPTION OF EXPERIMENT

Table 2 summarizes the description of each task, an agents performance with random policy, and the performance of the experts.

Table 2: Description of each task, an agents performance with random policy, and the performance of the experts.. $\mathbf{dim}(\mathcal{S})$ and $\mathbf{dim}(\mathcal{A})$ denote dimensionality of state and action spaces respectively.

| Task | $\mathbf{dim}(\mathcal{S})$ | $\mathbf{dim}(\mathcal{A})$ | Random Policy | Expert's Performance |
|---|---|---|---|---|
| Reacher-v1 | 11 | 2 | -43.21 ± 4.32 | -3.93 ± 1.58 |
| HalfCheetah-v1 | 17 | 6 | -282.43 ± 79.53 | 4130.22 ± 75.51 |
| Hopper-v1 | 11 | 3 | 14.47 ± 7.96 | 3778.05 ± 3.34 |
| Walker2d-v1 | 17 | 6 | 0.57 ± 4.59 | 5510.67 ± 74.44 |
| Ant-v1 | 111 | 8 | -69.68 ± 111.10 | 4812.93 ± 122.26 |
| Humanoid-v1 | 376 | 17 | 122.87 ± 35.11 | 10395.51 ± 205.81 |

## B    IMPLEMENTATION DETAILS

We implemented our algorithm with three neural networks each of which represents function $\mu_\theta$, $R_\omega$, and $V_\eta$. For convenience, we call the networks as deterministic policy network (DPN) for $\mu_\theta$, reward network (RN) for $R_\omega$, and state screening network (SSN) for $V_\eta$. All networks were two layer perceptrons that used leaky rectified nonlinearity (Maas et al., 2013a) except for the final output. DPN had 100 hidden units in each hidden layer, and its final output was followed by hyperbolic tangent nonlinearity to bound its action range. Note that the number of hidden units in DPN was the same as that of networks used in (Ho & Ermon, 2016). RN had 1000 hidden units in each hidden layer and a single output was followed by sigmoid nonlinearity to bound the output between [0,1]. The input of RN was given by concatenated vector representations for the state-action pairs. The SSN had the same architecture as RN had except for its input which was just vector representations for the state. The choice of hidden layer width in both RN and SSN was based on preliminary experimental results.

We employed a sampling strategy which was almost the same as deep deterministic policy gradient (DDPG) algorithm (Lillicrap et al., 2015) does. When updating RN and SSN, the state-action pairs were sampled from the replay buffers $\mathcal{B}_\beta$ and a buffer $\mathcal{B}_E$ that stores the datasets. When updating DPN, the states were sampled from the replay buffers $\mathcal{B}_\beta$.

We employed RMSProp (Hinton et al., 2012) for learning parameters of RN, DPN and SSN with a learning rate $10^{-4}$, a decay rate 0.99, and epsilon $10^{-10}$ for all tasks. The weights and biases in all layers were initialized by Xavier initialization (Maas et al., 2013b). We used coefficient $\epsilon = 10^{-2}$ for $v_\eta$, mini-batch size N = 64, size of the replay buffer $|\mathcal{B}_\beta| = 5 \times 10^5$. Regarding to the maximum number of episodes for training, we used M = 10000 for Humanoid-v1, and M = 2000 for the other tasks.

