# OpenReview forum: "Deterministic Policy Imitation Gradient Algorithm"
_ICLR.cc/2018/Conference — Reject_

### Official Review · AnonReviewer3 · 2017-11-26
**Hard to read**

**Rating:** 6
**Confidence:** 4

**Review:**

This paper proposes to extend the determinist policy gradient algorithm to learn from demonstrations. The method is combined with a type of density estimation of the expert to avoid noisy policy updates. It is tested on Mujoco tasks with expert demonstrations generated with a pre-trained network.

I found the paper a bit hard to read. My interpretation is that the main original contribution of the paper (besides changing a stochastic policy for a deterministic one) is to integrate an automatic estimate of the density of the expert (probability of a state to be visited by the expert policy) so that the policy is not updated by gradient coming from transitions that are unlikely to be generated by the expert policy.

I do think that this part is interesting and I would have liked this trick to be used with other imitation methods. Indeed, the deterministic policy is certainly helpful but it is tested in a deterministic continuous control task. So I'm not sure about how it generalizes to other tasks. Also, the expert demonstration are generated by the pre-trained network so the distribution of the expert is indeed the distribution of the optimal policy. So I'm not sure the experiments tell a lot. But if the density estimation could be combined with other methods and tested on other tasks, I think this could be a good paper.

---

> ### Author Response · Authors · 2018-01-05
> **Thank you for positive evaluations.**
>
> Thank you for your constructive comments and positive evaluations on our paper. We will clarify the role of SSF in the camera-ready version.
>
> > My interpretation is that the main original contribution of the paper (besides changing a stochastic policy for a deterministic one) is to integrate an automatic estimate of the density of the expert (probability of a state to be visited by the expert policy)
>
> Thank you for clearly understanding the role of SSF.
>
> > Indeed, the deterministic policy is certainly helpful but it is tested in a deterministic continuous control task. So I'm not sure about how it generalizes to other tasks.
>
> The expert's policy used in the experimetns is a stochastic one. Hence, the proposed method works not only on a deterministic continuous control tasks but also a stochastic one. We expect that it generalizes well to other tasks.

---

### Official Review · AnonReviewer1 · 2017-11-26
**This paper proposes an extension of the generative adversarial imitation learning (GAIL) algorithm by replacing the stochastic policy of the learner with a deterministic one. Simulation results with MuJoCo physics simulator show that this simple trick reduces the amount of needed data by an order of magnitude.**

**Rating:** 5
**Confidence:** 4

**Review:**

This paper considers the problem of model-free imitation learning. The problem is formulated in the framework of generative adversarial imitation learning (GAIL), wherein we alternate between optimizing reward parameters and learner policy's parameters. The reward parameters are optimized so that the margin between the cost of the learner's policy and the expert's policy is maximized. The learner's policy is optimized (using any model-free RL method) so that the same cost margin is minimized. Previous formulation of GAIL uses a stochastic behavior policy and the RIENFORCE-like algorithms. The authors of this paper propose to use a deterministic policy instead, and apply the deterministic policy gradient DPG (Silver et al., 2014) for optimizing the behavior policy.
The authors also briefly discuss the problem of the little overlap between the teacher's covered state space and the learner's. A state screening function (SSF) method is proposed to drive the learner to remain in areas of the state space that have been covered by the teacher. Although, a more detailed discussion and a clearer explanation is needed to clarify what SSF is actually doing, based on the provided formulation.
Except from a few typos here and there, the paper is overall well-written. The proposed idea seems new. However, the reviewer finds the main contribution rather incremental in its nature. Replacing a stochastic policy with a deterministic one does not change much the original GAIL algorithm, since the adoption of stochastic policies is often used just to have differentiable parameterized policies, and if the action space is continuous, then there is not much need for it (except for exploration, which is done here through re-initializations anyway). My guess is that if someone would use the GAIL algorithm for real problems (e.g, robotic task), they would significantly reduce the stochasticity of the behavior policy, which would make it virtually similar in term of data efficiency to the proposed method.
Pros:
- A new GAIL formulation for saving on interaction data.
Cons:
- Incremental improvement over GAIL
- Experiments only on simulated toy problems
- No theoretical guarantees for the state screening function (SSF) method

---

> ### Author Response · Authors · 2018-01-05
> **Responses**
>
> Thank you for your constructive comments on our paper. We will fix typos and clarify the role of SSF in the camera-ready version.
>
> > The authors also briefly discuss the problem of the little overlap between the teacher's covered state space and the learner's. A state screening function (SSF) method is proposed to drive the learner to remain in areas of the state space that have been covered by the teacher.
>
> The main purpose of introducing a SSF is not what you mentioned. Since we use the Jacobian of reward function to derive PG as opposed to prior IL works, the Jacobian is supposed to have information about how to get close to the expert's behavior for the learner. However, in the IRL objective (4), which is general in (max-margin) IRL literature, the reward function could know how the expert acts just only on the states appearing in the demonstration. In other words, the Jacobian could have information about how to get close to the expert's behavior just only on states appearing in the demonstration. What we claimed in Sec.3.2 is that the Jacobian for states which does not appear in the demonstration is just garbage for the learner since it does not give any information about how to get close to the expert. The main purpose of introducing the SSF is to sweep the garbage as much as possible.
>
> > However, the reviewer finds the main contribution rather incremental in its nature. Replacing a stochastic policy with a deterministic one does not change much the original GAIL algorithm, since the adoption of stochastic policies is often used just to have differentiable parameterized policies, and if the action space is continuous, then there is not much need for it (except for exploration, which is done here through re-initializations anyway)
>
> Figure.1 shows worse performance of Ours \setminus SSF which just replace a stochastic policy with a deterministic one. If Ours \setminus SSF worked well, we agree with your opinion that the main contribution is just incremental. However, introducing the SSF besides replacing a stochastic policy with a deterministic one is required to imitate the expert's behavior. Hence, we don't agree that the proposed method is just incremental.
>
> > My guess is that if someone would use the GAIL algorithm for real problems (e.g, robotic task), they would reduce the stochasticity of the behavior policy, which would make it virtually similar in term of data efficiency to the proposed method.
>
> Because the GAIL algorithm is an on-policy algorithm, it essentially requires much interactions for an update and never uses behavior policy. Hence, it would not make it virtually similar in term of data efficiency to the proposed method which is off-policy algorithm.
>
> > Cons:
> > - Incremental improvement over GAIL
>
> As mentioned above, we think that the proposed method is not just incremental improvement over GAIL.
>
> > - Experiments only on simulated toy problems
>
> We wonder why you thought the Mujoco tasks are just "toy" problems. Even though those tasks are not real-world problems, they have not been solved until GAIL has been proposed. In addition, the variants of GAIL (Baram et al., 2017; Wang et al., 2017; Hausman et al.) also evaluated their performance using those tasks. Hence, we think that those tasks are enough difficult to solve and can be used as a well-suited benchmark to evaluate whether the proposed method is applicable to the real-world problems in comparison with other IL algorithms.

---

### Official Review · AnonReviewer2 · 2017-11-27
**Combines IRL, adversarial training, and ideas from deterministic policy gradients. Paper is hard to read. MuJoCo results are good.**

**Rating:** 5
**Confidence:** 3

**Review:**

The paper lists 5 previous very recent papers that combine IRL, adversarial learning, and stochastic policies. The goal of this paper is to do the same thing but with deterministic policies as a way of decreasing the sample complexity. The approach is related to that used in the deterministic policy gradient work. Imitation learning results on the standard control problems appear very encouraging.

Detailed comments:

"s with environment" -> "s with the environment"?

"that IL algorithm" -> "that IL algorithms".

"e to the real-world environments" -> "e to real-world environments".

" two folds" -> " two fold".

"adopting deterministic policy" -> "adopting a deterministic policy".

"those appeared on the expert’s demonstrations" -> "those appearing in the expert’s demonstrations".

"t tens of times less interactions" -> "t tens of times fewer interactions".

Ok, I can't flag all of the examples of disfluency. The examples above come from just the abstract. The text of the paper seems even less well edited. I'd highly recommend getting some help proof reading the work.

"Thus, the noisy policy updates could frequently be performed in IL and make the learner’s policy poor. From this observation, we assume that preventing the noisy policy updates with states that are not typical of those appeared on the expert’s demonstrations benefits to the imitation.": The justification for filtering is pretty weak. What is the statistical basis for doing so? Is it a form of a standard variance reduction approach? Is it a novel variance reduction approach? If so, is it more generally applicable?

Unfortunately, the text in Figure 1 is too small. The smallest font size you should use is that of a footnote in the text. As such, it is very difficult to assess the results.

As best I can tell, the empirical results seem impressive and interesting.

---

> ### Author Response · Authors · 2018-01-05
> **Responses**
>
> Thank you for your constructive comments on our paper. We will fix typos and Figure.1. in the camera-ready version.
>
> > The justification for filtering is pretty weak.
>
> Since Figure.1 shows worse performance of Ours \setminus SSF which does not filter states appearing in the demonstration, we think that the justification is enough.
>
> > What is the statistical basis for doing so?
>
> Introducing a SSF is a kind of heuristic method, but it works as mentioned above.
>
> > Is it a form of a standard variance reduction approach? Is it a novel variance reduction approach? If so, is it more generally applicable?
>
> Introducing the SSF itself is not a variance reduction approach. We would say that direct use of the Joacobian of (single-step) reward function rather than that of Q-function to derive the PG (8) might reduce the variance because the range of outputs are bounded.
> Since we use the Jacobian of reward function to derive PG as opposed to prior IL works, the Jacobian is supposed to have information about how to get close to the expert's behavior for the learner. However, in the IRL objective (4), which is general in (max-margin) IRL literature, the reward function could know how the expert acts just only on the states appearing in the demonstration. In other words, the Jacobian could have the information about how to get close to the expert's behavior just only on states appearing in the demonstration. What we claimed in Sec.3.2 is that the Jacobian for states which does not appear in the demonstration is just garbage for the learner since it does not give any information about how to get close to the expert. The main purpose of introducing the SSF is to sweep the garbage as much as possible. The prior IL works have never mentioned about the garbage.

---

### Decision · Program_Chairs · 2018-01-29
**ICLR 2018 Conference Acceptance Decision**

**Decision:**

Reject

**Comment:**

All of the reviewers found some aspects of the formulation and experiments interesting, but they found the paper hard to read and understand. Some of the components of the technique such as the state screening function (SSF) seem ad-hoc and heuristic without much justification. Please improve the exposition and remove the unnecessary component of the technique, or come up with better justifications.